# CRL4^Cdt2^ Ubiquitin Ligase, A Genome Caretaker Controlled by Cdt2 Binding to PCNA and DNA

**DOI:** 10.3390/genes13020266

**Published:** 2022-01-29

**Authors:** Muadz Ahmad Mazian, Kumpei Yamanishi, Mohd Zulhilmi Abdul Rahman, Menega Ganasen, Hideo Nishitani

**Affiliations:** 1Faculty of Applied Science, Universiti Teknologi MARA, Cawangan Negeri Sembilan, Kampus Kuala Pilah, Kuala Pilah 72000, Negeri Sembilan, Malaysia; muadzam@uitm.edu.my; 2Applied Environmental Microbiology (EmiBio), Universiti Teknologi MARA, Cawangan Negeri Sembilan, Kampus Kuala Pilah, Kuala Pilah 72000, Negeri Sembilan, Malaysia; 3Graduate School of Science, University of Hyogo, Kamigori 678-1297, Japan; kunpeiy1991@gmail.com; 4Biotechnology Division, Centre for Cocoa Biotechnology Research, Malaysia Cocoa Board, Kota Kinabalu 88460, Sabah, Malaysia; hilmi@koko.gov.my; 5Department of Biological Sciences, National University of Singapore, Singapore 117558, Singapore

**Keywords:** CRL4^Cdt2^, cell cycle, ubiquitination, DNA binding domain, PIP box, intrinsically disordered region, IDR, CDK phosphorylation

## Abstract

The ubiquitin ligase CRL4^Cdt2^ plays a vital role in preserving genomic integrity by regulating essential proteins during S phase and after DNA damage. Deregulation of CRL4^Cdt2^ during the cell cycle can cause DNA re-replication, which correlates with malignant transformation and tumor growth. CRL4^Cdt2^ regulates a broad spectrum of cell cycle substrates for ubiquitination and proteolysis, including Cdc10-dependent transcript 1 or Chromatin licensing and DNA replication factor 1 (Cdt1), histone H4K20 mono-methyltransferase (Set8) and cyclin-dependent kinase inhibitor 1 (p21), which regulate DNA replication. However, the mechanism it operates via its substrate receptor, Cdc10-dependent transcript 2 (Cdt2), is not fully understood. This review describes the essential features of the N-terminal and C-terminal parts of Cdt2 that regulate CRL4 ubiquitination activity, including the substrate recognition domain, intrinsically disordered region (IDR), phosphorylation sites, the PCNA-interacting protein-box (PIP) box motif and the DNA binding domain. Drugs targeting these specific domains of Cdt2 could have potential for the treatment of cancer.

## 1. Introduction

The third principle of cell theory states that a cell arises from a pre-existing living cell through cell division [1]. Each cell divides by passing through a cell cycle, a sequence of specified stages, to produce two daughter cells. The cell cycle is divided into four stages: G1 phase, during which cells grow and prepare for DNA synthesis; S phase, during which DNA replicates; G2 phase, during which cells prepare for mitosis; and M phase, during which genetic material undergoes segregation and cells undergo cytokinesis [2]. The transmission of genetic information must be carefully regulated to assure that DNA is properly inherited by daughter cells, allowing these cells to function properly and survive. Cell cycle processes must be strictly regulated to prevent neoplasia [3], defined as the growth of malignant cells due to chromosomal instability, DNA damage and/or re-replication. 

Proteolysis plays a crucial role in properly regulating cell cycle progression. Several important proteolytic systems act during DNA replication to preserve genetic information [4,5,6]. Initiation of DNA replication in S phase activates the proliferating cell nuclear antigen (PCNA)-dependent proteolysis of licensing factor, Cdt1 by CRL4^Cdt2^, a Cullin RING E3 ubiquitin ligase, to inhibit re-replication. 

CRL4^Cdt2^ has been shown to be a master regulator of genome stability due to its ability to ubiquitinate various substrates, leading to their degradation through proteasome-mediated proteolysis, specifically during S phase and after DNA damage [7]. Cdt1 plays a vital role during the initiation of DNA replication, acting as a licensing factor during late M phase and early G1 phase [8,9,10]. Cdt1 cooperates with cell division cycle 6 (Cdc6) to aid in loading the hexameric minichromosome maintenance (MCM2-7) helicase complex at sites that bind origin recognition complex (ORC). This results in the formation of a pre-replicative complex (pre-RC) at origins of replication, which is essential for the initiation of DNA replication. The levels of Cdt1 increase during G1 phase but eventually decline as cells enter S phase [11,12]. The ubiquitin-mediated proteolytic degradation of Cdt1, Set8 and p21 by the CRL4 (DDB1-CUL4-Rbx1)-Cdt2 protein complex, hereafter referred to as CRL4^Cdt2^, is vital to ensure once-per-cell-cycle DNA replication, as depletion of Cdt2 induces massive re-replication [13]. Few studies, however, have evaluated the CRL4^Cdt2^ protein complex and its ubiquitination mechanism during S phase and after DNA damage. The lack of an X-ray crystal 3D structure of the Cdt2 protein has impeded understanding of the mechanism of action of CRL4^Cdt2^ ubiquitin substrates. This review, together with molecular modeling of Cdt2 domain structure, summarizes recent findings on the role of the C-terminal portion of Cdt2 in higher eukaryotes and provides insight into the mechanisms underlying substrate recognition and regulation. 

## 2. Cdt2- DDB1-CUL4-Rbx1 Complex, CRL4^Cdt2^

### 2.1. Cullin4 RING Ubiquitin Ligase, CRL4

Members of the Cullin RING ubiquitin ligase (CRL) family of E3 ubiquitin ligases play vital roles in regulating various cellular pathways [14,15]. To date, seven members of the CRL family, termed CRL1-7, have been identified in higher eukaryotes. The scaffold of one of these, CRL4, is based on the Cullin 4 subfamilies, Cullin 4A (CUL4A) and Cullin 4B (CUL4B). The N-terminal portion of CUL4 interacts with its substrate adapter DNA damage binding protein 1 (DDB1), which has three β-propeller or WD-40 repeat domains, BPA, BPB, and BPC domains. The BPB domain interacts with the N-terminal part of CUL4, whereas the BPA and BPC domains interact with their substrate receptors, referred to as DDB1-CUL4 associated factors (DCAFs), such as Cdt2 (Figure 1A) [7,13]. The activation of CRLs depends on the protein modification with NEDD8, called neddylation. The C-terminal portion of CUL4 conjugates with NEDD8 and is linked to the binding of the RING domain protein Rbx1, to which E2 is recruited [13]. 

### 2.2. Cdc10-Dependent Transcript-2, Cdt2

Cdt2 was initially isolated, together with Cdt1, as a Cdc10-dependent transcript-2 from fission yeast. Cdt2, also known as DCAF2 and DCX (DTL) [13,17], is one of the 90 DCAFs, substrate receptors associated with CRL4 ubiquitin ligase [7]. In contrast to Cdt2 from fission yeast, which mainly consists of WD40 repeats, Cdt2 proteins from higher eukaryotes contain more than 700 aa and have extended C-terminal regions (Figure 1B) [18,19]. However, the effect of the length of the Cdt2 C-terminus on CRL4 ligase activity remains unclear. The conserved N-terminal region, containing seven WD-40 repeats, forms a β-propeller structure (Figure 2A) [13,20]. The less conserved C-terminal region contains phosphorylation sites, the DNA binding domain (DBD), and the PCNA-interacting protein-box (PIP box) (refer to Section 4).

### 2.3. Overview of Substrate Recognition by CRL4^Cdt2^ (PCNA^DNA^-Dependent)

The active CRL4^Cdt2^ ubiquitin ligase requires a PCNA loaded onto DNA, or PCNA^DNA^, to recognize its substrates [21,22]. PCNA is a homotrimeric, ring-shaped replication clamp encircling DNA that aides in DNA replication and DNA damage repair. During S phase and upon DNA damage, PCNA is loaded onto chromatin, where it serves as a platform for proteins involved in DNA replication, DNA repair and chromatin metabolism to interact with each other [23,24].

CRL4^Cdt2^ target substrates possess a PIP degron for efficient ubiquitination (Figure 3). A PIP degron is a canonical PIP box with a TD motif, consisting of threonine and aspartate residues at the fifth and sixth positions of the PIP box, respectively, and a B+4 basic residue (K/R) at the fourth amino acid downstream of the PIP box for high-affinity PCNA binding [24,25,26]. In human cells, a basic residue (K/R) at the third position (B+3) is also important [25]. Following the binding of a PIP degron to PCNA^DNA^, these amino acids are required for their recognition by Cdt2 as a substrate-PCNA^DNA^ complex [22,25,26]. 

## 3. Substrate Recognition by the N-Terminal Region of Cdt2

Although no N-terminal Cdt2 structure has yet been described, it is thought to have a structure similar to that of another E3 ligase DCAF, DDB2. CRL4^DDB2^ is an E3 ligase that recognizes UV-photoproducts and promotes ubiquitination of the nucleotide excision repair factor XPC during UV-induced damage repair [27]. Analogous to the DDB2 structure [28], the N-terminal Cdt2 is predicted to interact with BPA and BPC WD40 repeats of DDB1. This interaction is thought to occur on one side, whereas the other side of the Cdt2 structure is thought to recognize the substrates for degradation (Figure 2 and Figure 4B) [7]. 

Upon binding to the p21 PIP degron, the surface charge of the interdomain connecting loop of PCNA, which connects two similar lobes of a PCNA monomer, changes from negative to positive [25]. This positive charge is thought to create an electrostatic charge, resulting in its recognition by the negatively charged surface of Cdt2. We have attempted to evaluate the substrate binding interaction at the N-terminal region of Cdt2 by simulating docking with ZDOCK [29]. Docking was performed between the modeled N-terminal structure of Cdt2 (amino acids 45–400) and the complex crystal structure of PCNA and a peptide derived from the C-terminal of p21 (139GRKRRQTSMTDFYHSKRRLIFS160) (PDB ID: 1AXC) [30]. The N-terminal region of Cdt2 (45–400) was modeled from a template of a highly repetitive propeller structure (PDB ID: 2YMU) using SWISS MODEL [31]. We performed a rigid body search in the 6D rotational and translational space at the default algorithms of ZDOCK version 3.0.2. Docking failed when no interacting residues were selected. Therefore, to improve docking accuracy, several previously described interacting residues of p21 (Q144, M147, T148, D149, F150, Y151, K154, and R155) were selected [30]. Docking revealed several interacting residues at the center of the Cdt2 β-propeller structure. Possible interactions between residues K144, R161, D381 and E362 of Cdt2 and residues S153, T148, D149, H152 and K154 (B+3) of p21 were observed (Figure 4C). Five amino acid residues (K144, R161, D381, E362 and K271) at the center of the Cdt2 β-propeller structure were predicted by the WDSPdb [32] database to be potential substrate interaction hotspots, confirming the docking results. Although the docking simulation found that residue K271 of Cdt2 did not directly interact with the substrate or PCNA, the mutation K271A resulted in the electrostatic potential at the center of the Cdt2 β-propeller structure being far more negatively charged than with wild-type Cdt2 (Figure 2D). Similarly, mutating all five residues to alanine markedly altered the electrostatic potential at the surface of the β-propeller structure (Figure 2D), which might affect the ability of Cdt2 to recognize and bind to its substrates on PCNA. Mutation analysis is required to confirm these predictions. We also found that residue R155 (B+4) on p21 interacted with residue E124 on PCNA, but not on Cdt2. These interactions confirm the importance of the TD amino acid motif and the B+3 and B+4 basic amino acids downstream of the PIP box in p21 for recognition by Cdt2 [25]. 

Although the simulation suggested that the Cdt2 N-terminal region alone was sufficient for recruitment to PIP-degron-bound PCNA^DNA^, UV-induced DNA damage showed that the Cdt2 N-terminal region (residues 1–417) alone was not recruited to this site [33]. The failed recruitment of Cdt2 resulted in the stabilization of the critical substrates Cdt1, Set8 and p21, which are prone to cause DNA re-replication [19,34,35]. Thus, the C-terminal of Cdt2 is required for this molecule to have a full ubiquitin ligase activity.

## 4. Essential Features in the C-Terminus of Cdt2 That Regulate CRL4^Cdt2^ Activity

The less conserved C-terminus of Cdt2 is associated with the intrinsically disordered region (IDR) of this molecule, with the pattern of the disordered region conserved in Cdt2 proteins from humans, mice, chickens, zebrafish, and *Xenopus* (Figure 1C). Furthermore, this IDR has been shown to play a crucial role in regulating Cdt2 activity [19,33,36]. Our group has successfully identified the essential features present in the C-terminus of Cdt2 that regulate Cdt2 activity [19,33,36].

### 4.1. PIP Box at the C-Terminal End of Cdt2

The end of the less conserved C-terminal region of Cdt2 has been found to contain a PCNA-interacting protein-motif or PIP box, whose amino acid sequence (Q-x-x-L/V/I/M-x-x-F/Y) (Figure 5A) is conserved in cells of all higher eukaryotes from Zebrafish to human [33,37,38]. The rapid degradation of Cdt2 substrates, such as Cdt1, Set8 and p21, coincides with the co-localization of Cdt2 with PCNA, in which Cdt2 is recruited through its C-terminal region to PCNA^DNA^ [24,33,39]. The interaction of this C-terminal region of Cdt2 with PCNA^DNA^ has been reported to be essential for the degradation of Xic1, a *Xenopus* ortholog of p21 in *Xenopus* egg extracts [21], and of Cdt1 and p21 in humans [33]. The recruitment of Cdt2 to PCNA^DNA^ was also observed in damaged chromatin following laser micro-irradiation and micropore UVC irradiation [33,39]. Recruitment of the C-terminal region of Cdt2 alone (residues 390–730) to PCNA^DNA^ after localized UVC irradiation was similar to that of full-length Cdt2. Conversely, recruitment of the N-terminal region of Cdt2 (residues 1–417) to PCNA^DNA^ was not observed, despite the N-terminal region being responsible for substrate recognition. These findings indicated that the PIP box in the C-terminus of Cdt2 is important for the recognition of PCNA^DNA^ and is required for the interaction of Cdt2 with PCNA. The affinity of Cdt2 PIP peptide for PCNA was found to be strong, similar to that of the p21 PIP peptide, and two orders of magnitude higher than that of Cdt1 [33]. The interaction of the PIP box with PCNA somehow triggers or assists in the binding of the N-terminus of Cdt2 to its substrates, thus allowing polyubiquitination-mediated proteolysis to occur. 

To confirm that the PIP box drives recruitment of Cdt2 to PCNA^DNA^, the PIP box motif was mutated, and recruitment was analyzed. Mutation of amino acids in the PIP box consensus sequence abrogated the recruitment of Cdt2 to PCNA^DNA^, resulting in less effective substrate ubiquitination [22,33,38].

### 4.2. Phosphorylation of the C-Terminal of Cdt2

Cyclin-dependent kinase (CDK)-mediated phosphorylation plays a vital role in tightly orchestrating cell cycle events for accurate transmission of genomic information [40]. Phosphorylation by CDKs is highly specific, as these enzymes only recognize the S/T-P consensus motif of target proteins [41,42,43,44], altering the function of regulators in vital processes, such as DNA replication and mitotic progression [42]. Human Cdt2 was reported to be a target for phosphorylation by CDKs (Figure 5B) because its extended C-terminal region contains 18 S/T-P sites [36]. 

Cyclins A/CDK2 and B/CDK1 have been reported to initiate the phosphorylation of Cdt2 during S and M phases, respectively [36,45]. CRL4^Cdt2^ has been found to co-localize with PCNA^DNA^ at the onset of S phase [33,36,38]. During late S phase, however, CRL4^Cdt2^ co-localization with PCNA^DNA^ was reduced, and re-accumulation of the CRL4^Cdt2^ substrates Set8 and thymine DNA glycosylase (TDG) was observed [45]. Re-accumulation of Cdt1 did not occur, however, because another ubiquitin ligase, CRL1^Skp2^, was also activated during this period, targeting Cdt1 for degradation [36]. Re-accumulation of CRL4^Cdt2^ substrates was associated with the hyperphosphorylation of Cdt2 because, both during G1 phase and the onset of S phase, low levels of phosphorylated Cdt2 can be detected [36,46]. 

The CDK phosphorylation at the S/T-P sites of the C-terminal of Cdt2 [19,36] has been reported to reduce the ability of Cdt2 to bind to PCNA^DNA^ during late S phase, reducing CRL4^Cdt2^ ubiquitination activity [13]. Substrate re-accumulation has been reported to correlate with Cdt2 hyperphosphorylation [36]. Stably expressed Cdt2 with mutations at all 18 phosphorylation sites (Cdt2-18A mutant) showed reductions in phosphorylation, when compared with wild-type Cdt2, both in vivo and in vitro. Interestingly, expression of the Cdt2-18A mutant, thus abrogating Cdt2 phosphorylation, was found to prevent the re-accumulation of CRL4^Cdt2^ substrates from late S phase to G2 phase, with the same result observed when cells expressing wild-type Cdt2 were treated with a CDK1 inhibitor [45]. Cdt2 is hyperphosphorylated during M phase. Following UV irradiation, Cdt2 substrates were not degraded during M phase, although PCNA was loaded onto DNA [45,47]. Similarly, Cdt2-18A was found to induce the degradation of its substrate during M phase after UV irradiation [36]. 

Phosphorylation of PCNA-interacting proteins, such as DNA polymerase delta p68 subunit, replication factor C subunit 1 (RFC1), DNA ligase I and flap endonuclease 1 (Fen1), has been reported to regulate their interactions with PCNA [48,49,50,51]. The phosphorylation levels of DNA ligase I and Fen1 have been found to increase toward late S or G2 phase, and their affinity to PCNA decreased similar to that of Cdt2 [48,50]. Phosphorylation of p68 subunit and RFC1 also inhibited their interactions with PCNA [49,51], which may direct the disassembly of the replication machinery at the end of S phase. CDK, which is primarily associated with cyclin A, is the key kinase responsible for the phosphorylation of these proteins [48,51]. However, the mechanism by which phosphorylation reduces their affinity for PCNA is not well understood. Phosphorylation within the PIP box of DNA pol delta p68 subunit has been reported to inhibit its interaction with PCNA [49]. In other proteins, however, inhibitory phosphorylation sites are located at positions from 50 to several hundred amino acids removed from the PIP box [48,50,51]. CDK phosphorylation of Cdt2 close to the PIP box can reduce its affinity to PCNA. However, 18 CDK phosphorylation sites are scattered in the Cdt2 C-terminal region (Figure 5B), and the essential phosphorylation sites remain to be determined.

### 4.3. DNA Binding Domain in the C-Terminal Region

Cdt2 has been reported as unable to recognize its substrates that bind to PCNA unless PCNA is loaded onto DNA [13]. Questions remain regarding the relationship between Cdt2 activation and PCNA^DNA^, and about the mechanism by which PCNA loaded on DNA activates CRL4^Cdt2^ to couple its activity to DNA synthesis. The dependence of Cdt2 activity on S phase progression and DNA damage suggests a close relationship with DNA. 

Regions encompassing amino acid residues 471–570 and 570–690 in the C-terminal region of human Cdt2 were thought able to bind DNA; however, an in vitro assay showed that a peptide (amino acids 460–580) had DNA binding activity [19] (Figure 5C). Interestingly, the DNA binding domain (DBD) patterns in both regions were conserved in Cdt2 from higher eukaryotes (Chicken, Mouse and *Xenopus*) [19] (Figure 5C). Compared with the N-terminal alone (amino acids 1–417), the Cdt2 sequences 1–650, containing the DBD, retained high Cdt1 degradation activity, suggesting that binding to DNA increases CRL4^Cdt2^ ubiquitin ligase activity and couples CRL4^Cdt2^ activation to PCNA^DNA^ during S phase and after DNA damage [13,19]. The structural stability of Cdt2 may depend on bipartite binding via PIP-box binding to PCNA and DBD binding to DNA. Upon binding to DNA, the DBD can trigger alterations in Cdt2 structure. The kinetics of UV-induced Cdt1 degradation were found to be delayed in U2OS cells when DBD was replaced by a linker peptide [19]. Delayed cell cycle progression was also observed as an increase in the numbers of cells in G2/M phases when Cdt2 replaced with linker peptide was expressed. A similar pattern of delayed degradation and cell cycle progression was observed in cells expressing Cdt2 with a mutated PIP box. As shown by cells expressing only the N-terminus of Cdt2, losing both motifs (DBD and PIP box), suppressed the degradation of Cdt1 and induced re-replication of DNA and activation of the G2/M checkpoint. 

Interactions of DBD with DNA can be regulated through phosphorylation. Phosphorylation of DBD inhibited its interaction with DNA [52,53,54]. Hyperphosphorylation mainly occurs at the C-terminus of Cdt2, with DBD containing a cluster of CDK phosphorylation sites (Figure 5B). Hyperphosphorylation, especially of the DBD, may interfere with its binding to DNA, causing Cdt2 to dissociate from DNA. Mutations at phosphorylation sites in the C-terminal region of Cdt2 (Cdt2 18-A) have been consistently shown to increase its affinity for PCNA^DNA^ [36]. Furthermore, the phosphorylation-associated accumulation of negative charge from phosphate groups on DBD may result in electrostatic repulsion between DNA and Cdt2 (Figure 6). To date, however, the specific amino acids in the DBD whose phosphorylation suppress DNA binding have not been well characterized. It is therefore important to clarify the mechanism by which PIP-box binding to PCNA and DBD binding to DNA are mutually linked and co-regulated through phosphorylation.

In vitro DNA binding assays suggested that the C-terminus of Cdt2 has a preference for single-stranded DNA (ssDNA) over double-stranded DNA (dsDNA) [19]. The ssDNA is formed at the replication forks and the sites of DNA repair. Because PCNA is loaded at double strand-single strand junctions [23], Cdt2 can bind to both PCNA and ssDNA at the same time. 

## 5. Intrinsically Disordered Region in the C-Terminal Half of Cdt2 

Most of the Cdt2 C-terminus is thought to consist of a disordered region [19] (Figure 1). Under physiological conditions, intrinsically disordered regions (IDRs) in proteins are long unstructured or unfolded segments that do not form defined secondary and/or tertiary structures. IDRs, which are found in many proteins, interact with molecules such as proteins, DNA and RNA to initiate cell signaling [55,56]. IDRs contain short linear motifs (SLiMs), defined as peptide sequences of around 2 to 10 residues with biological activities [57]. These SLiMs can affect the interactions of IDRs with other molecules, thereby modulating their function [58,59]. The disordered region of Cdt2 contains multiple [S/T-P] CDK phosphorylation sites [36], a PIP box [33], and a predicted cyclin-binding motif [553RrL], along with a DNA-binding domain [19], suggesting that Cdt2 possesses an IDR with SLiMs. Many IDRs within proteins undergo transitions to ordered structures upon protein binding to partner molecules, as reported for the CDK inhibitors, p21 and p27, and transcription factors [56,60]. Post-translational modifications of IDRs, by, for example, phosphorylation and acetylation, modulate their function by altering their binding affinity or structural folding [59]. The DNA binding domain within IDR of Cdt2 is rich in positively charged amino acids (Figure 5C). 

Interestingly, the properties of the substrate Cdt1 are similar to those of Cdt2. The N-terminal region of Cdt1 is intrinsically disordered [61], with a PIP degron at the N-terminal end. The other substrates, Set8 and p21, also have PIP degrons in their IDRs [62], as many PIP motifs are detected in IDRs [63]. In addition, IDRs in mouse and *Drosophila* Cdt1 can bind DNA [61,64]. 

Several hypotheses have been proposed to explain the mechanisms by which these diverse properties are integrated to regulate CRL4^Cdt2^ activity universally. One hypothesis has proposed that a PIP degron in an IDR can help CRL4^Cdt2^ detect and bind to the degron-peptide exposed on PCNA because binding would be hindered if the PIP degron was located within the folded domain. A second hypothesis has proposed that the presence of a DNA binding domain with PIP motifs may help Cdt2 associate with both PCNA and DNA at the same time and orient the N-terminal substrate recognition domain in a suitable direction to bind to and ubiquitinate PCNA-bound substrates, thereby activating CRL4^Cdt2^ when PCNA is on DNA. The interdependence of PCNA and DNA binding may also constitute a fail-safe mechanism to ensure that only proteins that bind to PCNA^DNA^ are targeted for destruction. Some PCNA-binding proteins, including chromatin assembly factor 1 (CAF1), spartan (SPRTN) and budding yeast MutS homolog 6 (MSH6), also contain DNA binding domains [65,66,67]. The presence of a DNA binding domain along with a PIP box may constitute a universal mechanism, ensuring that PCNA-interacting proteins function only when they bind to PCNA^DNA^. A third hypothesis is based on the ability of IDRs or low complexity domains in solution to coalesce with each other and undergo liquid–liquid phase separation (LLPS) [68,69]. LLPS is a process of condensed membrane-less assembly, similar to nucleoli and stress granules, that is initiated by multivalent protein–protein and/or protein–nucleic acid interactions. This leads to the formation of condensed granule-like particles when a critical concentration of molecules is reached. Details of this mechanism remain unclear, but IDRs or low complexity domains are more likely to interact with each other due to their biased amino acid composition and unstructured features [69,70]. ORC, Cdc6, and Cdt1 were recently found to contain IDRs that co-assemble and undergo LLPS, assisting MCM2-7 recruitment and its loading onto replication origins [61,71]. LLPS by these factors is enhanced in the presence of DNA but is inhibited by the presence of CDK. Similarly, we speculate IDR proteins with PIP-boxes in the replication factory assemble around PCNA^DNA^ when replication forks are created. This assembly would be enhanced when IDR proteins undergo an LLPS-like transition promoted by the presence of DNA. During this assembly, the DNA-binding domain of Cdt2 could accelerate CRL4^Cdt2^-mediated ubiquitination. Conversely, phosphorylation of Cdt2 by CDK could inhibit its binding to DNA and its removal from LLPS, thereby reducing CRL4^Cdt2^ activity. Similarly, the phosphorylated forms of Cdt1 may be excluded from undergoing LLPS and become refractory to degradation [72]. Biochemical analyses are required to confirm this hypothesis.

## 6. Conclusions and Perspective

The detailed structure-function analysis of Cdt2 can provide insights into the regulation of CRL4^Cdt2^ ubiquitination activity through its PIP box, phosphorylation sites, and DNA binding domain (Figure 6). The IDR of Cdt2, located in its C-terminal region, is key to orchestrating the interactions of Cdt2 with other proteins, ensuring a proper cell cycle. The neddylation inhibitor pevonedistat (MLN4924), which induces re-replication and cell death [73], is currently being tested as a promising treatment for cancer. This drug inhibits all members of the Cullin family of ubiquitin ligases that require neddylation for their activity. Because some tumor cells express high levels of Cdt2 protein [74,75], Cdt2-specific targeting reagents may be more promising anti-cancer drugs. The arrangement of important features within the IDR and hotspot residues required for substrate recognition could act as significant targets for drugs that interfere with the role of specific domains to cure cancer. 

## Figures and Tables

**Figure 1 genes-13-00266-f001:**
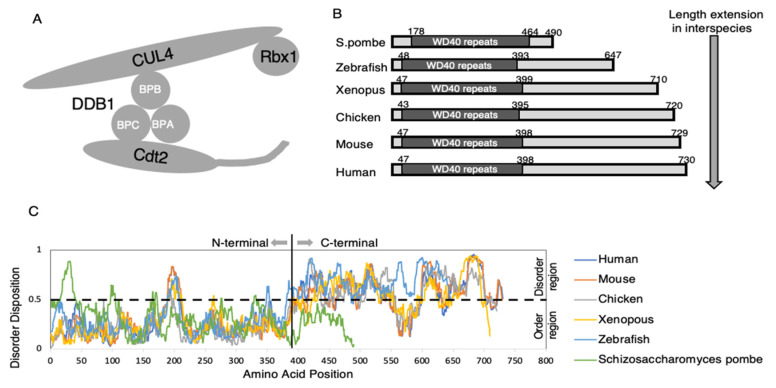
Schematic diagram of CRL4^Cdt2^, comparison of the extended C-terminal regions of Cdt2 from various species, and dynamic prediction analyses of the N-terminal and C-terminal portions of Cdt2 protein. (**A**) Schematic representation of the CRL4^Cdt2^ complex. Shown are the substrate receptor Cdt2, the adapter protein DDB1 (BPA, BPC, BPC), a scaffold protein, CUL4, and Rbx1. (**B**) Comparison of extended C-terminal regions of Cdt2 from yeast to human. (**C**) Dynamic prediction analyses of the N-terminal and C-terminal portions of the Cdt2 substrate receptor protein. The flexible patterns in the disordered region are conserved in higher eukaryotes. Dynamic prediction analysis was performed using Predictor of Natural Disorder Regions (PONDR^®^VLXT) software [16].

**Figure 2 genes-13-00266-f002:**
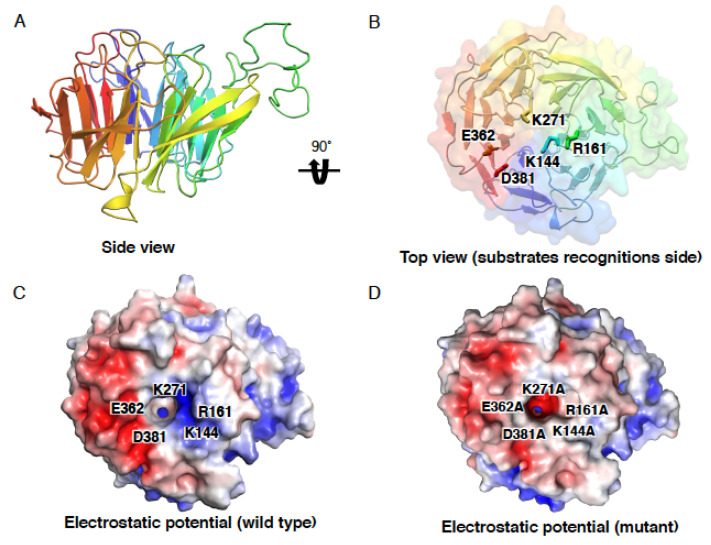
Predicted N-terminal structure of Cdt2 (residues 45–400). (**A**) Highly repetitive residue 45–400, modeled using SWISS MODEL (PDB ID: 2YMU). (**B**) Top view of the β-propeller structure. Hotspot residues that may be involved in substrate recognition are shown. (**C**) Electrostatic potential of the wild-type β-propeller structure illustrated in (**B**). (**D**) Electrostatic potential of the β-propeller structure containing five mutations. The surface electrostatic potential was found to be highly altered by these mutations, especially at residue K271 in the binding cavity.

**Figure 3 genes-13-00266-f003:**
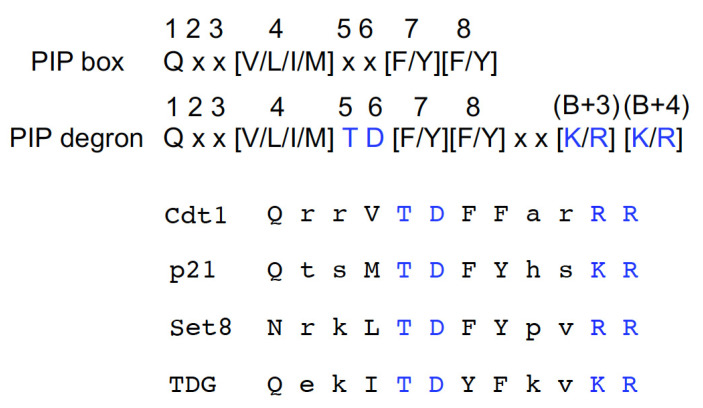
PIP box and PIP degron amino acid sequences. The TD motif (threonine and aspartate residues) at the fifth and sixth positions and the basic amino acids [K/R] at the third and fourth positions downstream of the PIP box are crucial for substrate recognition and degradation. The PIP degrons found in representative substrates are listed.

**Figure 4 genes-13-00266-f004:**
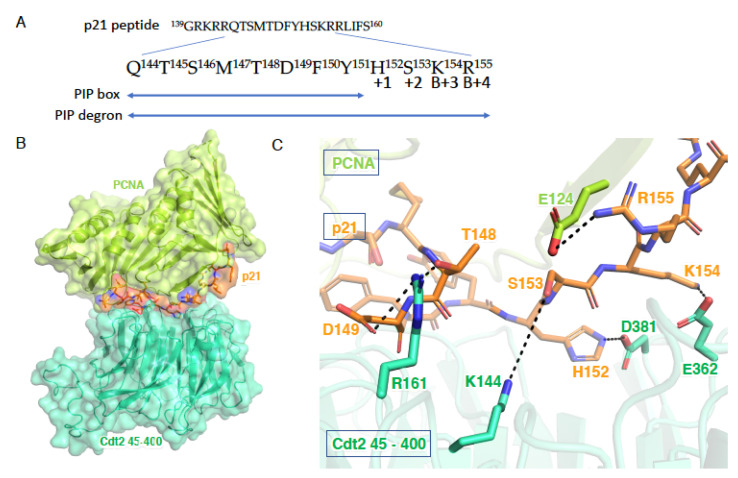
Simulation of the docking of a PCNA-p21 complex with the N-terminal region (residues 45–400) of Cdt2. (**A**) Amino acid sequences of the p21 PIP box and PIP degron. (**B**) Schematic showing the docking in solution of the PCNA-p21 complex (PDB ID: 1AXC) with the N-terminal structure Cdt2. (**C**) Schematic showing the possible hydrogen bonding of residues R161, K144, D381, and E362 of Cdt2 with the p21 peptide and residue R155 of p21 with E124 on PCNA.

**Figure 5 genes-13-00266-f005:**
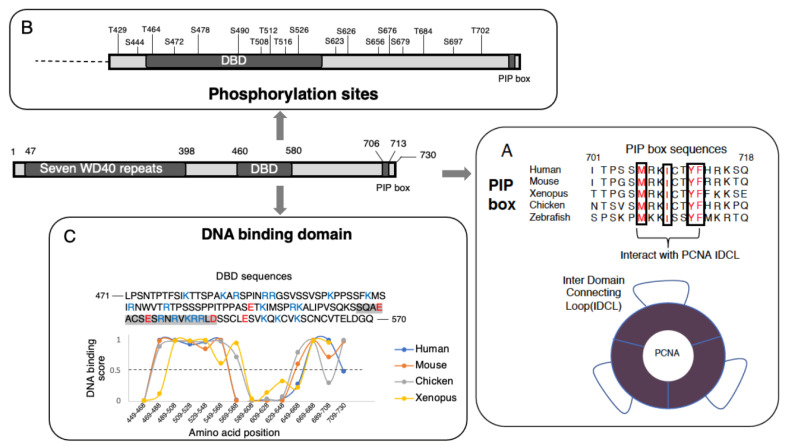
Illustration of the C-terminal region of human Cdt2. (**A**) PIP box motif conserved among higher eukaryotes. The red-colored amino acids have been observed to interact with the interdomain connecting loop (IDL) and its underneath hydrophobic pocket of PCNA. (**B**) Eighteen CDK-phosphorylation sites along the C-terminal region of Cdt2. (**C**) DNA binding domain prediction (DNABIND) analysis of Cdt2 [19], showing a highly conserved DNA binding pattern in the C-terminal region of Cdt2 (amino acids 460–580) of higher eukaryotes. The amino acids shown in blue (K/R) and red (D/E) letters are positively and negatively charged, respectively. The highlighted amino acids (SQAEACSESRNRVKRRLD) are predicted to form a helix-turn-helix structure.

**Figure 6 genes-13-00266-f006:**
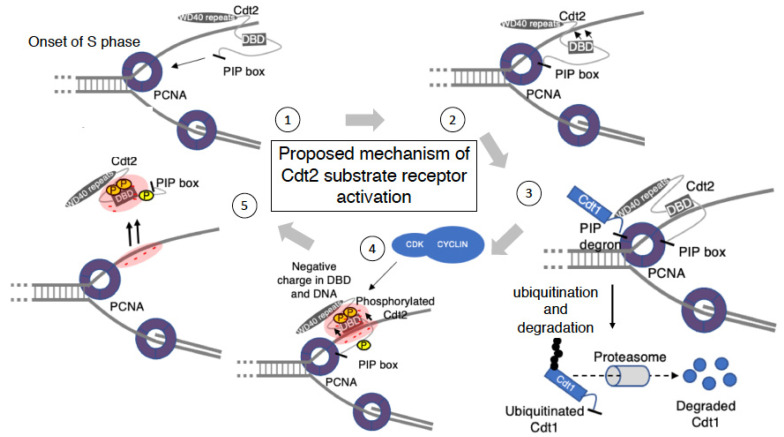
Proposed mechanism of Cdt2 substrate receptor activation (**1**). Interaction of Cdt2 with PCNA^DNA^, driven by the PIP box motif at the C-terminal end of Cdt2. (**2**). Interaction between the DNA binding domain (DBD) and single-stranded ssDNA to provide a stable and robust anchor for Cdt2 during substrate “catching”. (**3**). Recruitment of a substrate (Cdt1) independent of CRL4^Cdt2^ through its PIP box. Cdt2 recognizes the PIP degron of Cdt1, as described in detail in Section 2 and Section 3, for ubiquitination and degradation by the proteasome. (**4**). Phosphorylation of Cdt2 by the CDK-Cylin complex at the end of S phase. The accumulation of negative charge from phosphate groups triggers electrostatic repulsion between Cdt2 and DNA and a reduction of PIP box affinity to PCNA^DNA^, which together lead to dissociation from PCNA, by an as yet unknown mechanism. (**5**). Detachment of phosphorylated Cdt2 from and its inability to bind to PCNA^DNA^, abrogating its ubiquitination activity. In the case of DNA damage, a similar process from step 1–3 is initiated.

## Data Availability

The data presented in this study are available on request from the corresponding authors.

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
