# Peer review of "CRL4Cdt2 Ubiquitin Ligase, A Genome Caretaker Controlled by Cdt2 Binding to PCNA and DNA"

_genes, 2022, doi:10.3390/genes13020266_

Round 1

Reviewer 1 Report

The authors present a review about the interaction between Cdt2 and the CRL4 ligase and how this interaction is regulated by DNA binding. The review covers a relevant topic in DNA repair. It is well-written and covers the topic exhaustively, including molecular modelling results where structural biology data has not yet been reported. Insufficiently documented molecular docking exercises are given substantial space in this review. These results are very likely inaccurate and should be corrected. 

Major issues:

1.) Molecular modelling results presented in reviews: The authors report results from two molecular docking simulations, 1.) PCNA-p21 complex and the N-terminal region of Cdt2 as well as 2.) C-terminal region of Cdt2 and ssDNA. While the results of both docking exercises are presented with the expected caution in the figure legends ("possible" in Figure 3B, and 5A), the figures nevertheless show atomic resolution structural data with interatomic distances plotted to 0.1A accuracy. The use of all information sources available for a review, including databases and molecular models is commendable, however the difficulty that arises here is that - in my opinion - reporting extensive molecular modelling in a review, which usually lacks a materials and methods part, introduces statements on molecular structure and function into the literature that can not be reproduced, since the details are not reported. I think that this issue could be addressed by introduction of a brief methods section that explains in detail how the docking experiments were conducted, either as separated section of the review or by introducing the relevant information in the main text or figure legends.

2.) Molecular modelling: The authors use the protein modeller provided by Swiss-Prot to model the N-terminal Cdt2 structure. Rather than the model with the highest likelihood, the one modelled using 5NNZ, they chose the second highest ranked model, 2YMU, for reasons not stated. Protein structure prediction has recently advanced dramatically with the advent of Alphafold. From my own experience and the experience of my colleagues so far, Alphafold models are of exceptional quality and very likely to exceed any model prepared with a traditional modelling method in quality. Comparison of the SWISS MODEL model and the Alphafold model shows overall highly similar structure, but an RMSD of 3.7A between the models suggests that there is a certain level of uncertainty with respect to the precision of the position of the side chains (as depicted in Fig 3B).

3.) Docking 1 (PCNA-p21 complex and the N-terminal region of Cdt2): The authors mention that they have used ZDOCK to perform the molecular docking simulation. ZDOCK should be referenced (PMID: 24532726) and the version of ZDOCK used should be stated. Furthermore, it would be helpful to briefly mention how ZDOCK operates, i.e. that it performs rigid body docking, taking into account electrostatics and shape, but to my knowledge e.g. not solvent. This approach to docking often works, but it's far from being a real molecular dynamics simulation. I assume that WDSP refers to the http://www.wdspdb.com/wdsp/ website, but here it again would be essential to include a reference to the original paper, which might be PMID: 22916195. What was the input, what was the output? What kind of analysis was provided by WDSP?

4.) Docking 2 (C-terminal region of Cdt2 and ssDNA): Why was Cdt2 modelled with Galaxyweb? A reference to GalaxyWeb is missing. When the entire Cdt2 protein is modelled in Alphafold, no alpha helices are predicted for the regions shown in Fig 5. Instead, almost every part of the protein outside of the WD40 propeller is expected to be unstructured, with an exception at the N-terminus. The protein structure prediction plot shown in Fig. 1C corroborates this result. Given that the docking between this model and DNA is analyzed to great detail in Figure 5, the provenience of the Cdt2 model should be made more clear. Reasoning why this part of the C-terminus is believed to be helical should be presented. Is there biophysical data in the literature, e.g. circular dichroism data, to show that it's helical? How was the structure of the ssDNA determined? In the figure it looks like the authors used essentially half a dsDNA (repeat distance around 10), but there is no evidence that ssDNA adopts this structure in solution; I would actually consider this highly improbable. For the docking itself, no information on the method is provided. How was the docking carried out? Which algorithm, which server? Please add the references to the used software. If there is little evidence that Cdt2 C-terminus is helical and if the model for ssDNA is simply half a double-helix and the fitting is performed as rigid body fit, what, in the view of the authors, is the probability that the result is meaningful? Again, I commend that the authors try to extract as much information from the available data as possible for their review and I really appreciate it. However, the approach shown in Fig. 5 is insufficient and should be replaced by a properly documented molecular dynamics simulation, which should be feasible with the size of the binding partners and would add valuable information.

Minor issues:

5.) Protein structure prediction: Which program was used to generate the results shown in Fig. 1C. Please add the information to the figure legend and add a citation for the software.

6.) Acronyms should be resolved the first time they appear. If I am not mistaken, there is an option to add a glossary for this purpose in MDPI journals. Could be considered here.

Typographic errors:

The review is overall well-written and - in my, non-native speaker, opinion - does not require extensive language editing. However, there are some minor issues that need to be addressed.

2: Anatomy .. presents; Title reads a bit awkward. Maybe consider to re-phrase.

17: a -> the

20: the mechanism .. is not fully understood.

24: to cure cancer diseases -> to treat cancer.

32: presence of essential proteins involved is inquired for cell’s decision to replicate... Please consider rephrasing.

46: acronym of MCM should be explained here where it first occurs. disassembly -> disassemble

49: vast -> a vast number

54: comma before which

59/60: To date, how CRL4Cdt2 and its ubiquitination mechanism activate during the cell cycle precisely in the S phase and after DNA damage is not completely understood. 

60: lack of a 

63: in-silico insight based on Cdt2 63 prediction structure. Please consider rephrasing and adding more detail.

67: 'prominent' in this context is maybe a bit too anthropocentric. How about "The Cullin RING ubiquitin ligases (CRLs) family of E3 ubiquitin ligases has a vital role in regulating diverse cellular pathways."

80: It's not really a pathway for complex formation. "Schematic representation of CRL4Cdt2 complex."

91: "the evolutionary length" Please consider rephrasing to make it more clear.

111/112: Unclear, please consider rephrasing or adding context.

117: DDB2 (capitalization)

126/127: was changed -> changes

127: conversed -> inverted?

151: a UV induced...

162: citation missing

176: Please consider rephrasing.

191/192: Please consider rephrasing.

233 What does the "ex." mean?

245 ...the essential phosphorylation sites remain to be determined.

247: not -> unless 

249: Please rephrase.

251: Please rephrase.

262/263: Please rephrase.

272: Please rephrase.

282: is -> are

308/309/310: Please rephrase.

317 ; -> such as

320/321: Reference not clear, please rephrase. 

335: s PIP-degron

339: it -> them

341-343: Please rephrase.

344: "assemble each other" do you mean into a protein structure? If it's into a LLPS, assembly might not be the best word, as there's no increase in structure or complexity. Maybe "coalesce" would be a good word for this case?

374: Müller-wille -> Müller-Wille

Author Response

Comments and Suggestions for Authors

The authors present a review about the interaction between Cdt2 and the CRL4 ligase and how this interaction is regulated by DNA binding. The review covers a relevant topic in DNA repair. It is well-written and covers the topic exhaustively, including molecular modelling results where structural biology data has not yet been reported. Insufficiently documented molecular docking exercises are given substantial space in this review. These results are very likely inaccurate and should be corrected. 

We would like to thank Reviewer 1 for critical reading of our Review article and instructive comments and suggestions. The helped improve our paper very much.

Major issues:

1.) Molecular modelling results presented in reviews: The authors report results from two molecular docking simulations, 1.) PCNA-p21 complex and the N-terminal region of Cdt2 as well as 2.) C-terminal region of Cdt2 and ssDNA. While the results of both docking exercises are presented with the expected caution in the figure legends ("possible" in Figure 3B, and 5A), the figures nevertheless show atomic resolution structural data with interatomic distances plotted to 0.1A accuracy.

Reply: We agree that it is not accurate to show the interatomic distances plotted to 0.1A precision in Figures 3 and 5.

Correction: We have removed the interatomic distances labeled in Figures 3 and Figure 5 was deleted in the revised manuscript.

1.) The use of all information sources available for a review, including databases and molecular models is commendable, however the difficulty that arises here is that - in my opinion - reporting extensive molecular modelling in a review, which usually lacks a materials and methods part, introduces statements on molecular structure and function into the literature that can not be reproduced, since the details are not reported. I think that this issue could be addressed by introduction of a brief methods section that explains in detail how the docking experiments were conducted, either as separated section of the review or by introducing the relevant information in the main text or figure legends.

We fully agree with the reviewer’s comments and a brief methodology of the docking simulation have been added in the main text.

A brief methodology for the N-Terminal docking was included in the text,

We have attempted to evaluate the substrate binding interaction at the N-terminal region of Cdt2 by simulating docking with ZDOCK [29]. Docking was performed between the modeled N-terminal structure of Cdt2 (amino acids 45-400) and the complex crystal structure of PCNA and a peptide derived from the C-terminal of p21 (139GRKRRQTSMTDFYHSKRRLIFS160) (PDB ID: 1AXC) [30]. The N-terminal region of Cdt2 (45-400) was modeled from a template of a highly repetitive propeller structure (PDB ID: 2YMU) using SWISS MODEL [31]. We performed a rigid body search in the 6D rotational and translational space at the default algorithms of ZDOCK version 3.0.2. Docking failed when no interacting residues were selected. Therefore, to improve docking accuracy, several previously described interacting residues of p21 (Q144, M147, T148, D149, F150, Y151, K154, and R155) were selected [30]. Docking revealed several interacting residues at the center of the Cdt2 β-propeller structure. Possible interactions between residues K144, R161, D381 and E362 of Cdt2 and residues S153, T148, D149, H152 and K154 (B+3) of p21 were observed (Figure 3C). Five amino acid residues (K144, R161, D381, E362 and K271) at the center of the Cdt2 β-propeller structure were predicted by the WDSPdb [32]database to be potential substrate interaction hotspots, confirming the docking results. Although the docking simulation found that residue K271 of Cdt2 did not directly interact with the substrate or PCNA, the mutation K271A resulted in the electrostatic potential at the center of the Cdt2 β-propeller structure being far more negatively charged than with wild-type Cdt2 (Figure 2D). Similarly, mutating all five residues to alanine markedly altered the electrostatic potential at the surface of the β-propeller structure (Figure 2D), which might affect the ability of Cdt2 to recognize and bind to its substrates on PCNA. Mutation analysis is required to confirm these predictions. We also found that residue R155 (B+4) on p21 interacted with residue E124 on PCNA, but not on Cdt2. These interactions confirm the importance of the TD amino acid motif and the B+3 and B+4 basic amino acids downstream of the PIP box in p21 for recognition by Cdt2 [25].

DBD docking brief methodology

“"The docking experiments were conducted by using YASARA docking tool, an AutoDock based tool for molecular docking (included within licensed YASARA STRUCTURE version 12.5.7, YASARA Biosciences GmbH, Austria) in which the C-terminal region of Cdt2 was treated as the receptor and the modelled ssDNA structure (based on dsDNA structure from PDB ID:1BNA) was treated as the ligand. Two local docking experiments were conducted with two different docking grid setups (i) by creating a specific docking grid to encompass residues 471-555 and 10Å radius around them (ii) docking grid that encompassed residues 535-560 and 10Å radius around them. A total of 25 docking runs were conducted for each setup and the highest-ranking docked protein–DNA complexes were analyzed for the comparative binding energies and dissociation constant (Kd) of the docked molecular complexes."

However, we strongly agree with the reviewer that this docking simulation is highly questionable. Hence, we would like to delete this section from the revised manuscript.

2.) Molecular modelling: The authors use the protein modeller provided by Swiss-Prot to model the N-terminal Cdt2 structure. Rather than the model with the highest likelihood, the one modelled using 5NNZ, they chose the second highest ranked model, 2YMU, for reasons not stated. Protein structure prediction has recently advanced dramatically with the advent of Alphafold. From my own experience and the experience of my colleagues so far, Alphafold models are of exceptional quality and very likely to exceed any model prepared with a traditional modelling method in quality. Comparison of the SWISS MODEL model and the Alphafold model shows overall highly similar structure, but an RMSD of 3.7A between the models suggests that there is a certain level of uncertainty with respect to the precision of the position of the side

chains (as depicted in Fig 3B).

The N-terminal modeling was performed by our colleague. Back then 2YMU was the highest template that available. The template 5NNZ was available recently.

As in the figure 3B (now 3C), we agreed with the reviewer and we remove the distance to avoid any confusion of the interaction

3.) Docking 1 (PCNA-p21 complex and the N-terminal region of Cdt2): The authors mention that they have used ZDOCK to perform the molecular docking simulation. ZDOCK should be referenced (PMID: 24532726) and the version of ZDOCK used should be stated. Furthermore, it would be helpful to briefly mention how ZDOCK operates, i.e. that it performs rigid body docking, taking into account electrostatics and shape, but to my knowledge e.g. not solvent. This approach to docking often works, but it's far from being a real molecular dynamics simulation. I assume that WDSP refers to the http://www.wdspdb.com/wdsp/ website, but here it again would be essential to include a reference to the original paper, which might be PMID: 22916195. What was the input, what was the output? What kind of analysis was provided by WDSP?

Thank you for your concern. We have updated in the text the ZDOCK version used (with citation) and the analysis provided by WDSP. (Please refer to text answer in question 1)

4.) How was the structure of the ssDNA determined? In the figure it looks like the authors used essentially half a dsDNA (repeat distance around 10), but there is no evidence that ssDNA adopts this structure in solution; I would actually consider this highly improbable. (Question 4)

Based on the experimental data, the DNA binding assay for the C-terminal domain was conducted using calf thymus DNA (ctDNA). However, since there is no availability of ctDNA structure in PDB, we chose DNA structure from Protein Data Bank (PDB ID:1BNA) as our reference DNA structure. Then we removed one strand of the dsDNA structure and performed energy minimization experiment. The resulting ssDNA structure was then energy minimized and was treated as the ligand for the subsequent docking experiment.

Anyway, we deleted this section from the revised manuscript.

4.) If there is little evidence that Cdt2 C-terminus is helical and if the model for ssDNA is simply half a double-helix and the fitting is performed as rigid body fit, what, in the view of the authors, is the probability that the result is meaningful? (Question 4)

The results of these experiments will help us to determine the possible interaction between the proposed DNA binding sites of C-terminus and ssDNA and will guide us in the future mutagenesis studies to find out the hotspot or the key residues that initiate the DNA binding.

We deleted this section from the revised manuscript.

4.) Again, I commend that the authors try to extract as much information from the available data as possible for their review and I really appreciate it. However, the approach shown in Fig. 5 is insufficient and should be replaced by a properly documented molecular dynamics simulation, which should be feasible with the size of the binding partners and would add valuable information

Thank you for the suggestion on the molecular dynamics simulation and we will take full consideration of this matter in our future studies. However, for this manuscript, it is not suitable for us to conduct intensive molecular dynamics simulation studies as the focus of this current manuscript is as a review article and not full research paper.

Thank you for the comments. As the reviewer pointed out, when the structure of the overall Cdt2 or the truncated C-terminal region predicted by any modelling software other than Galaxyweb, no secondary structure was predicted for the selected region. In contrast, prediction using Galaxyweb resulted in a helix-turn-helix (HTH) structure. Unfortunately for now, we don’t have any biophysical data to support this prediction. We were highly interested in determining the possible DNA binding residues or the hotspot residue that might play a key role in initiating the DNA binding. Since the Galaxyweb prediction showed HTH domain, which is a major structural motif that is capable of binding to DNA, we decided to proceed with the docking simulation. Although we were unable to obtain a docking model with high accuracy, we wanted to report the possible DNA interacting residues from the docking models, which can be useful for mutagenesis studies in future. However, we strongly agree with the reviewer that this docking simulation is highly questionable. Hence, we would like to remove this section from the manuscript.

Minor issues:

5.) Protein structure prediction: Which program was used to generate the results shown in Fig. 1C. Please add the information to the figure legend and add a citation for the software.

The information of software used and citation was added in the text

“……Dynamic prediction analysis was performed using Predictor of Natural Disorder Regions (PONDR®VLXT) software [16].

6.) Acronyms should be resolved the first time they appear. If I am not mistaken, there is an option to add a glossary for this purpose in MDPI journals. Could be considered here.

Thank you for your suggestion. We really appreciate it.

Typographic errors:

The review is overall well-written and - in my, non-native speaker, opinion - does not require extensive language editing. However, there are some minor issues that need to be addressed.

2: Anatomy .. presents; Title reads a bit awkward. Maybe consider to re-phrase.

Diversity And Universality; Anatomy Of Cdt2 Present A Fine Control Of Ubiquitination Activity Of CRL4 Ligase On DNA

We changed the tittle to:

CRL4Cdt2 ubiquitin ligase; a genome caretaker controlled by Cdt2 binding to PCNA and DNA

17: a -> the

Corrected to “the”

20: the mechanism .. is not fully understood.

Corrected to “However, the mechanism it operates via its substrate receptor, Cdt2, is not fully understood.”

24: to cure cancer diseases -> to treat cancer.

Corrected to “Drugs targeting these specific domains of Cdt2 could have potential for the treatment of cancer.” in a revised text

32: presence of essential proteins involved is inquired for cell’s decision to replicate... Please consider rephrasing.

We rephrased, “G1 phase, during which cells grow and prepare for DNA synthesis;”

46: acronym of MCM should be explained here where it first occurs. disassembly -> disassemble

Mini chromosome maintenance (MCM2-7) Line 49.

Disassemble (the sentence containing this word was removed from the text)

49: vast -> a vast number

various

54: comma before which

“,”

59/60: To date, how CRL4Cdt2 and its ubiquitination mechanism activate during the cell cycle precisely in the S phase and after DNA damage is not completely understood

Rephrased to;

“Few studies, however, have evaluated the CRL4Cdt2 protein complex and its ubiquitination mechanism during S phase and after DNA damage.”

60: lack of a 

“The lack of X-ray” was corrected to “lack of an X-ray”

63: in-silico insight based on Cdt2 63 prediction structure. Please consider rephrasing and adding more detail.

Rephrased to:

This review, based on molecular modeling of Cdt2 domain structure, summarizes recent findings on the role of the C-terminal portion of Cdt2 in higher eukaryotes and provides insight into the mechanisms underlying substrate recognition and regulation.

67: 'prominent' in this context is maybe a bit too anthropocentric. How about "The Cullin RING ubiquitin ligases (CRLs) family of E3 ubiquitin ligases has a vital role in regulating diverse cellular pathways."

Thank you for a nice suggestion.

Rephrased to;

Members of the Cullin RING ubiquitin ligase (CRL) family of E3 ubiquitin ligases play vital roles in regulating various cellular pathways

80: It's not really a pathway for complex formation. "Schematic representation of CRL4Cdt2 complex."

“CRL4Cdt2 complex formation” was corrected to “Schematic representation of CRL4Cdt2 complex."

91: "the evolutionary length" Please consider rephrasing to make it more clear.

We removed the “evolutionary” in the text and Figure 1B.

111/112: Unclear, please consider rephrasing or adding context.

To make clear, we added Table 1 to show the amino acid sequence of PIP-degron and the examples of proteins possessing PIP-degron, and quoted them as follows,

“CRL4Cdt2 target substrates possess a PIP degron for efficient ubiquitination (Table 1). A PIP degron is a canonical PIP box with a TD motif, consisting of threonine and aspartate residues at the fifth and sixth positions of the PIP box, respectively, and a B+4 basic residue (K/R) at the fourth amino acid downstream of the PIP box for high-affinity PCNA binding [24,25,26]. In human cells, a basic residue (K/R) at the third position (B+3) is also important [25]. Following the binding of a PIP degron to PCNADNA, these amino acids are required for their recognition by Cdt2 as a substrate-PCNADNA complex [22,25,26].”

(+1)

Table 1

117: DDB2 (capitalization)

Corrected to “DDB2”

126/127: was changed -> changes

Corrected to “changes”

127: conversed -> inverted?

Corrected to

“This positive charge is thought to create an electrostatic charge, resulting in its recognition by the negatively charged surface of Cdt2.”

151: a UV induced...

Corrected to “a UV induced…”

162: citation missing

References [19,33,36] are quoted.

176: Please consider rephrasing.

Rephrased to

“Recruitment of the C-terminal region of Cdt2 alone (residues 390-730) to PCNADNA after localized UVC irradiation was similar to that of full-length Cdt2.”

191/192: Please consider rephrasing.

Rephrased to

“The amino acids shown in blue (K/R) and red (D/E) letters are positively and negatively charged, respectively.”

233 What does the "ex." mean?

Corrected to “such as”

245 ...the essential phosphorylation sites remain to be determined.

As suggested, corrected to

“the essential phosphorylation sites remain to be determined.”

247: not -> unless 

Corrected to “unless”

249: Please rephrase.

251: Please rephrase.

Lines 249-251 were corrected;

“Questions remain regarding the relationship between Cdt2 activation and PCNADNA, and about the mechanism by which PCNA loaded on DNA activates CRL4Cdt2 to couple its activity to DNA synthesis. The dependence of Cdt2 activity on S phase progression and DNA damage suggests a close relationship with DNA.”

262/263: Please rephrase.

Until the Cdt2 structure is solved, it is worth saying that perhaps the DBD triggers the structure altering of Cdt2 when it binds to DNA

was changed to

“Upon binding to DNA, the DBD can trigger alterations in Cdt2 structure.”

272: Please rephrase.

Rephrased to:

Interactions of DBD with DNA can be regulated through phosphorylation.

282: is -> are

Rephrased to,

“To date, however, the specific amino acids in the DBD whose phosphorylation suppress DNA binding have not been well characterized.”

308/309/310: Please rephrase.

This part was removed in the revised manuscript.

317 ; -> such as

“such as” was added.

320/321: Reference not clear, please rephrase. 

As the Cdt2 C-terminal part contains multiple CDK phosphorylation sites, PIP-box and DNA-binding domain, Cdt2 is assumed as a protein with IDR and SLiMs within it.

Rephrase to and References were included;

“The disordered region of Cdt2 contains multiple [S/T-P] CDK phosphorylation sites [36], a PIP box[33], and a predicted cyclin-binding motif [553RrL], along with a DNA-binding domain[19], suggesting that Cdt2 possesses an IDR with SLiMs.”

335: a PIP-degron

Corrected to;

 a PIP-degron

339: it -> them

and

341-343: Please rephrase.

Rephrased to,

“the presence of a DNA binding domain with PIP motifs may help Cdt2 associate with both PCNA and DNA at the same time and orient the N-terminal substrate recognition domain in a suitable direction to bind to and ubiquitinate PCNA-bound substrates, thereby activating CRL4Cdt2 when PCNA is on DNA.”

344: "assemble each other" do you mean into a protein structure? If it's into a LLPS, assembly might not be the best word, as there's no increase in structure or complexity. Maybe "coalesce" would be a good word for this case?

“assemble” was corrected to “coalesce”

374: Müller-wille -> Müller-Wille

corrected to “Wille”

Reviewer 2 Report

Dear authors,

your review on the function and regulation of Cdt2 present a rounded and interesting story.

Here are my main comments aiming to improve the accessibility of the text for a wider readership.

Title: I would like to suggest a shorter titel (articles with shorter titles have a higher probability od being read and downloaded) highlighting the key message; may be something along this line “Control of CRL4-dependent Ubiqitinylation by Cdt2 while bound to PCNA and DNA”

While the text needs some English editing, I will only highlight the important points I noticed.

L40 “recent work” not “works” better “ recent evidence”

L42 proteolysis of S phase licensing…

L43 – 47 I suggest to delete this section as it does not add important information to the story

L49 various substrates (not vast substrates)

L50 delete “in a cell cycle”

L52 the hexameric

L55 initiate DNA replication (delete the)

L55 The levels of Cdt1 increase (delete “were observed to”) – avoid indirect speech  as it weakens the impact od scientific writing

L56 decline better than degraded

L57 introduce Set8 and p21 briefly to the reader here

L57 ..proteolysis by the CRL4-Cdt2 protein complex thereafter referred to as CRL4Cdt2

L59 To date, research on the CRL4Cdt2 protein complex and its ubiquitinylation mechanism in S phase and after DNA damage is still sketchy.

L65 better The Cdt2-DDB1-Cul4-RBX1 complex (CRL4Cdt2)

L72 & L81 BPA, BPB and BPC domains (otherwise there is the danger that the domains are misread as proteins)

L98 please include the PDB-ID of the underlying structure used in SwissModel. Please explain the significance of the indicated amino acids in panel C

L118 Cdt2 is predicted to better than will

I wonder whether it may work better to swop section 3 and 4. Although the discussion would deal first with the C-terminal section of Cdt2, it would make sense from the point of the mechanism as the C-terminal section binds first before the N-terminal part engages the substrate.

L126-127 this sentence comes out of the blue and needs an introduction

L129 …its negative charged binding site” needs some context as unclear

L131 Please include a reference foe ZDOCK and WDSP

L133 might interact better than have an interaction

L143 please relate this information back to the amino acids mentioned in L140-141

L144 K271 of which protein? (please ensure that all amino acids are clearly assigned to their protein. For example, you could use a code like K271Cdt2-superscript)

I suggest to swop Figures 5 and 6 as the information relating to the currently figure 5 comes in the text after the information relating to figure 6

L211 as it contains 18 S/T-P sites in its extended C-terminal region

L227-228 this sentence needs context and explanation

L233 e.g. better than ex.

L249-251 The dependency of Cdt2 activity on S phase progression and DNA damage suggest a close relationship with DNA may be better.

L264 the interdependence of PCNA and DNA binding looks like a fail-save mechanism to ensure that only proteins bound to PCNA at the DNA are targeted.

Figure 6 and text: what remains unanswered between the lines is whether Cdt2 binds to the SAME PCNA ring where the targets like Cdt1 or p21 are or whether Cdt2 binds to a SECOND PCNA rind near by – may be worth mentioning in the text

L297 et al: it may help to mention that ssDNA accumulates at stalled forks and sites of DNA repair

L300 reference for GalaxyWeb

L304: R550, K552 and R553 of which protein?

L318 are short peptide sequences

Are the PIP box and the DNA binding domain SLIMS? Please clarify

L345 please introduce LLPS to the reader

Author Response

Comments and Suggestions for Authors

Dear authors,

your review on the function and regulation of Cdt2 present a rounded and interesting story.

Here are my main comments aiming to improve the accessibility of the text for a wider readership.

Title: I would like to suggest a shorter title (articles with shorter titles have a higher probability of being read and downloaded) highlighting the key message; may be something along this line “Control of CRL4-dependent Ubiqitinylation by Cdt2 while bound to PCNA and DNA”

We would like to thank Reviewer 2 for valuable and instructive comments and suggestions.

As suggested, we changed the title as follows,

CRL4Cdt2 ubiquitin ligase, a genome caretaker controlled by Cdt2 binding to PCNA and DNA

While the text needs some English editing, I will only highlight the important points I noticed. L40 “recent work” not “works” better “recent evidence”

“recent evidence” was removed and rephrased to,

“Proteolysis plays a crucial role in properly regulating cell cycle progression.”

L42 proteolysis of S phase licensing…

Rephrased to,

“Initiation of DNA replication in S phase activates the proliferating cell nuclear antigen (PCNA)-dependent proteolysis of licensing factor Cdt1 by CRL4Cdt

L43 – 47 I suggest to delete this section as it does not add important information to the story

Proteolysis plays a crucial role to regulate proper cell cycle progression.

The section was deleted as suggested.

L49 various substrates (not vast substrates)

Corrected to “various substrate

L50 delete “in a cell cycle”

We deleted “in a cell cycle”

L52 the hexameric

Corrected to “the hexameric..”

L55 initiate DNA replication (delete the)

Rephrased to,

“which is essential for the initiation of DNA replication.”

L55 The levels of Cdt1 increase (delete “were observed to”) – avoid indirect speech as it weakens the impact of scientific writing

Thank you for valuable comment.

We corrected, “The levels of Cdt1 increase”

L56 decline better than degraded

“but eventually decline as cells entered the S phase”

L57 introduce Set8 and p21 briefly to the reader here

Brief enzymatic functions were mentioned as follows,

“The degradation of Cdt1, Set8 (a histone H4K20 mono-methyltransferase that controls histone modifications) and p21(a cyclin-dependent kinase inhibitor)”

L57 ..proteolysis by the CRL4-Cdt2 protein complex thereafter referred to as CRL4Cdt2

Rephrased to,

“the CRL4 (DDB1-CUL4-Rbx1)-Cdt2 protein complex, hereafter referred to as CRL4Cdt22

L59 To date, research on the CRL4Cdt2 protein complex and its ubiquitinylation mechanism in S phase and after DNA damage is still sketchy.

Rephrased to,

“Few studies, however, have evaluated the CRL4Cdt2 protein complex and its ubiquitination mechanism during S phase and after DNA damage.”

L65 better The Cdt2-DDB1-Cul4-RBX1 complex (CRL4Cdt2)

Corrected to,

Cdt2- DDB1-CUL4-Rbx1 complex, CRL4Cdt2

L72 & L81 BPA, BPB and BPC domains (otherwise there is the danger that the domains are misread as proteins)

Corrected to “BPA, BPB and BPC domains”

L98 please include the PDB-ID of the underlying structure used in SwissModel. Please explain the significance of the indicated amino acids in panel

PDB-ID was included,

“Figure 2. Predicted N-terminal structure of Cdt2 (residues 45-400). (A) Highly repetitive residue 45-400, modeled using SWISS MODEL (PDB ID: 2YMU). “

The indicated hotspot amino acids were explained as follows,

“Five amino acid residues (K144, R161, D381, E362 and K271) at the center of the Cdt2 β-propeller structure were predicted by the WDSPdb [32]database to be potential substrate interaction hotspots, confirming the docking results.”

L118 Cdt2 is predicted to better than will

“the N-terminal Cdt2 is predicted to interact with BPA and BPC

Although the discussion would deal first with the C-terminal section of Cdt2, it would make sense from the point of the mechanism as the C-terminal section binds first before the N-terminal part engages the substrate.

We thank this reviewer for instructive suggestion. In the current form, we described the role of N-terminal region of Cdt2 as a substrate receptor at first, followed by C-terminal part. Since we introduced an overview of substrate recognition by CRL4-Cdt2 in the previous section 2.3, we think that readers will at first expect how the PIP-degron on PCNA was recognized by the substrate-receptor domain of Cdt2.Therefore, we would like to leave the current order.  

L126-127 this sentence comes out of the blue and needs an introduction

As responded to the Editor’s comment,

“After binding to the p21 PIP degron, the surface charge of the interdomain connecting loop of PCNA, which connects two similar lobes of a PCNA monomer, changes from negative to positive [25].”

L129 …its negative charged binding site” needs some context as unclear

Rephrased to,

“This positive charge is thought to create an electrostatic charge, resulting in its recognition by the negatively charged surface of Cdt2.”

L131 Please include a reference foe ZDOCK and WDSP

The reference for ZDOCK and WDSP added in the text as follws,

“docking with ZDOCK [29].” and “predicted by the WDSPdb [32] database”

L133 might interact better than have an interaction

Rephrased to,

“to be potential substrate interaction hotspots”

L143 please relate this information back to the amino acids mentioned in L140-141

To clearly relate the indicated amino acid residues, we labeled the amino acid residue numbers and listed as a new Figure 3A, together with an amino acid sequence of PIP-degron in Table 1.

L144 K271 of which protein? (please ensure that all amino acids are clearly assigned to their protein. For example, you could use a code like K271Cdt2-superscript)

We indicated the protein name as follows,

In the docking simulation, K271 of Cdt2 did not directly interact with the substrate or PCNA.

I suggest to swop Figures 5 and 6 as the information relating to the currently figure 5 comes in the text after the information relating to figure 6

We deleted Figure 6 together with the section of f DNA binding modeling, in respond to reviewer 1. Too much speculation might cause puzzling in the future.

L211 as it contains 18 S/T-P sites in its extended C-terminal region

Human Cdt2 was reported to be a phosphorylation target for CDKs (Figure 4B), as its extended C-terminal region contains 18 S/T-P sites[36].

L227-228 this sentence needs context and explanation

Rephrased as follows,

Cdt2 is hyperphosphorylated during M phase. Following UV irradiation, Cdt2 substrates were not degraded during M phase, although PCNA was loaded onto DNA.[45][47] Similarly, Cdt2-18A was found to induce the degradation of its substrate during M phase after UV irradiation [36].

L233 e.g. better than ex.

Corrected to “such as”

L249-251 The dependency of Cdt2 activity on S phase progression and DNA damage suggest a close relationship with DNA may be better.

This paragraph was corrected to,

“Cdt2 has been reported unable to recognize its substrates that bind to PCNA unless PCNA is loaded onto DNA [13]. Questions remain regarding the relationship between Cdt2 activation and PCNADNA, and about the mechanism by which PCNA loaded on DNA activates CRL4Cdt2 to couple its activity to DNA synthesis. The dependence of Cdt2 activity on S phase progression and DNA damage suggests a close relationship with DNA.”

L264 the interdependence of PCNA and DNA binding looks like a fail-save mechanism to ensure that only proteins bound to PCNA at the DNA are targeted.

Thank you for a valuable idea, we cited this idea in L347.

“The interdependence of PCNA and DNA binding may also constitute a fail-safe mechanism to ensure that only proteins that bind to PCNADNA are targeted for destruction.”

L297 et al: it may help to mention that ssDNA accumulates at stalled forks and sites of DNA repair

In vitro DNA binding assay suggested that the C-terminus of Cdt2 has a preference for ssDNA over dsDNA [18]. The ssDNA is formed at the replication forks and the sites of DNA repair. Because PCNA is loaded at double strand-single strand junctions [23], Cdt2 can bind to both PCNA and ssDNA at the same time.

L300 reference for GalaxyWeb

The modeling part of DNA binding domain was deleted in the revised manuscript.

L304: R550, K552 and R553 of which protein?

between R550, K552 and R553 of Cdt2 with the DNA bases

The modeling part of DNA binding domain was deleted in the revised manuscript.

L318 are short peptide sequences

A short introduction of SLiM was mentioned.

“short linear motifs (SLiMs), defined as peptide sequences of around 2-10 residues with biological activities [57].”

Are the PIP box and the DNA binding domain SLIMS? Please clarify

PIP box is normally eight aa lengths, thus can be a member of SLiM. We also included a predicted cyclin-binding motif [553RrL], along with a DNA-binding domain[19].

L345 please introduce LLPS to the reader

We mentioned LLPS in more detail from lines 352 to 359 of revised text,

“A third hypothesis is based on the ability of IDRs or low complexity domains in solution to coalesce with each other and undergo liquid-liquid phase separation (LLPS) [68,69]. LLPS is a process of condensed membrane-less assembly, similar to nucleoli and stress granules, that is initiated by multivalent protein-protein and/or protein-nucleic acid interactions. This leads to the formation of condensed granule-like particles when a critical concentration of molecules is reached. Details of this mechanism remain unclear , but IDRs or low complexity domains are more likely to interact with each other due to their biased amino acid composition and unstructured features [69,70]”

Round 2

Reviewer 1 Report

I would like to thank the authors for the extensive reworking of their manuscript and the detailed rebuttal letter. I find that their review is substantially improved now.

There are two minor typographic errors that need to be addressed:

127 paragraph heading should be bold

305 detailin -> detail in